# A Novel Data-Driven Evaluation Framework for Fork after Withholding Attack in Blockchain Systems

**DOI:** 10.3390/s22239125

**Published:** 2022-11-24

**Authors:** Yang Zhang, Yourong Chen, Kelei Miao, Tiaojuan Ren, Changchun Yang, Meng Han

**Affiliations:** 1College of Information Science and Technology, Zhejiang Shuren University, Hangzhou 310015, China; 2School of Computer Science and Artificial Intelligence, Changzhou University, Changzhou 213164, China; 3Binjiang Institute of Zhejiang University, Hangzhou 310053, China; 4Zhejiang University, Hangzhou 310058, China

**Keywords:** blockchain, faw attack, evaluation method, revenue model, computing power allocation optimization

## Abstract

In the blockchain system, mining pools are popular for miners to work collectively and obtain more revenue. Nowadays, there are consensus attacks that threaten the efficiency and security of mining pools. As a new type of consensus attack, the Fork After Withholding (FAW) attack can cause huge economic losses to mining pools. Currently, there are a few evaluation tools for FAW attacks, but it is still difficult to evaluate the FAW attack protection capability of target mining pools. To address the above problem, this paper proposes a novel evaluation framework for FAW attack protection of the target mining pools in blockchain systems. In this framework, we establish the revenue model for mining pools, including honest consensus revenue, block withholding revenue, successful fork revenue, and consensus cost. We also establish the revenue functions of target mining pools and other mining pools, respectively. In particular, we propose an efficient computing power allocation optimization algorithm (CPAOA) for FAW attacks against multiple target mining pools. We propose a model-solving algorithm based on improved Aquila optimization by improving the selection mechanism in different optimization stages, which can increase the convergence speed of the model solution and help find the optimal solution in computing power allocation. Furthermore, to greatly reduce the possibility of falling into local optimal solutions, we propose a solution update mechanism that combines the idea of scout bees in an artificial bee colony optimization algorithm and the constraint of allocating computing power. The experimental results show that the framework can effectively evaluate the revenue of various mining pools. CPAOA can quickly and accurately allocate the computing power of FAW attacks according to the computing power of the target mining pool. Thus, the proposed evaluation framework can effectively help evaluate the FAW attack protection capability of multiple target mining pools and ensure the security of the blockchain system.

## 1. Introduction

With the advancement of digitization, the data security problem of centralized intelligent systems has occurred (e.g., data theft, privacy theft, and illegal operations) resulting from the centralized processing of massive data [1,2,3]. Due to the characteristics of decentralization, time-series data, collective maintenance, and immutability of blockchain technology, the security requirements for intelligent systems have promoted the development of blockchain technology [4]. Nowadays, blockchain technology is widely applied in the Internet of Things, smart healthcare, smart commerce, and so on [5,6,7]. It is a crucial technology to speed up the development of digital industrialization; blockchain technology considerably enhances the efficiency and security of the digital industry. At present, academia and industry have devoted their efforts to researching the strategic frontier technology of blockchain and applying it to digital reform [8].

The consensus algorithm is one of the core technologies of blockchain, which can realize the consistency of massive network nodes on transactions [9]. Proof of Work (PoW) is one of the most widely used consensus algorithms. It has been applied to validate transactions and records of cryptocurrencies such as Bitcoin [10]. In the process of PoW, miners are required to continuously solve mathematical puzzles with their computing power to win the bookkeeping rights of blocks. The more blocks there are in the entire network, the harder mathematical puzzles to be solved by the miners will become. Thus, the probability that a single miner receives the revenue with its computing power will become lower, and a single miner may need to wait for a long time to receive the revenue. To receive more consistent revenue, some miners may choose to form mining pools. Within such pools, the pool manager concentrates all miners to reach a consensus and distribute the revenue in proportion to each miner’s contribution. Although mining pools provide miners with more stable revenue, there may be consensus attacks on mining pools during the consensus process [11], such as the Selfish Mining (SM) attack [12], Block Withholding (BWH) attack [13], and Fork After Withholding (FAW) attack. Consensus attacks affect the normal transactions of mining pools and provide transaction support for criminal acts such as money laundering, cyber extortion, and so on [14]. In the SM attack, the attacker does not broadcast the block after mining. When other miners have calculated the block, the attacker will immediately broadcast the mined block and compete for the current block revenue, which results in a fork in the blockchain network, i.e., the chain is forked into two branches [15]. However, the successful launch of the SM attacks depends on the high computing power and great block propagation efficiency of the attacker. The BWH attacker allocates its computing power. It uses part of the computing power for normal consensus and uses the remaining computing power to penetrate the target mining pool. Then only part of the PoW is reported, and the full PoW is discarded [16]. The computing power for penetration does not contribute to the target mining pools but still helps the attacker receive the revenue distributed by the target mining pool. However, BWH attacks may make mining pools fall into the miner’s dilemma problem. The FAW attack is a novel attack combining SM attacks and BWH attacks [17]. In the FAW attack, the attacker allocates its computing power in the same way as in the BWH attacks. It uses part of the computing power for launching penetration attacks. But the attacker does not discard the full PoW immediately and still keeps the block, as in SM attacks. When miners attacked by FAW know that other miners have calculated and mined, they will immediately announce the mined block and broadcast, thereby generating a fork to gain the revenue. The FAW attack makes the BWH attack’s revenue as the lower limit revenue and does not result in the miner’s dilemma problem. Therefore, the FAW attack can help the attacker obtain revenue from BWH attacks, then fork and reduce the revenue of the target mining pool. In conclusion, the FAW attack is a more threatening method against mining pools, which reduces the enthusiasm for mining pool consensus and endangers consensus security.

It is of great significance to study and evaluate the protection capability of the target mining pools against FAW attacks, which can improve the security of the target mining pools in the network. There are a few results in the area of evaluating FAW attacks at present. The main problems of traditional FAW attack evaluation methods are as follows:At present, there are a few evaluation frameworks for FAW attacks, and there is a lack of revenue evaluation when the target mining pool encounters FAW attacks. Effective security evaluation of the target mining pool is difficult.The evaluation of FAW attacks in the evaluation mining pool and target mining pool are both considered to be single, i.e., one evaluation mining pool evaluates one target mining pool. There is a lack of one evaluation mining pool that evaluates multiple target mining pools, and a consensus revenue model for evaluating the attack revenues of multiple target mining pools is also needed.The traditional FAW attack evaluation allocates the attack computing power fixedly, and excessive computing power may be wasted on the target mining pools with small revenue. Thus, the evaluation computing power cannot be efficiently allocated by the size of the target mining pool to maximize the attack revenue.Given the above problems, this paper considers the target mining pools to be multiple and proposes a novel evaluation framework for target mining pools. In this framework, we take advantage of optimization theory and the traditional FAW attack method and propose a computing power allocation optimization algorithm (CPAOA) of the FAW evaluation mining pool for multiple target mining pools. The main contributions of this paper are as follows:We propose a novel evaluation framework for target mining pools. We extend the evaluation of FAW attacks on one target mining pool to multiple target mining pools. We establish the mathematical equations for honest consensus revenue, block withholding revenue, successful fork revenue, consensus cost, etc. Then we establish an adaptive revenue optimization model for the computing power of the evaluation mining pools. At the same time, we also establish the revenue models of target mining pools and other mining pools to evaluate the attack revenue after the adaptive computing power allocation.We propose an improved Aquila optimization-based model-solving algorithm. More specifically, based on the traditional Aquila optimization algorithm, we improve the selection mechanism of different optimization processes and the fitness calculation of the proposed revenue optimization model. The algorithm can improve the convergence speed of the solution of the attack revenue optimization model and find the global optimal solution efficiently.We propose a solution update mechanism to prevent the algorithm from falling into local optimal solutions. The mechanism combines the idea of scout bees in an artificial bee colony optimization algorithm and the constraint of allocating computing power. According to the similarity ratio of solutions in the process of multiple iterations, the solution update mechanism can replace the relevant solutions in time and help prevent the algorithm from falling into the local optimal solution dilemma during the solution process.

## 2. Related Work

Currently, some scholars focus on using reinforcement learning, optimization, strategy selection, and other methods to study BWH attacks or SM attacks. For example, Fan et al. [18] proposed an adaptive zero-determinant strategy, which calculates the cooperative strategy revenue and betrayal strategy by calculating the cooperative strategy. The revenue method solves the problem of mutual BWH attacks between mining pools and non-cooperative mining. Davidson et al. [19] proposed a variable difficulty chains-based SM strategy. This strategy is generalized to different mining difficulties and the importance of making block timestamp manipulation a new component of the strategy space for selfish miners. The current relevant references can better implement BWH attacks or SM attacks, but BWH attacks will cause mining pools to fall into the miner’s dilemma in practice. The successful launch of an SM attack requires the attacker’s high computing power and fast block propagation efficiency. Moreover, they both have limitations in the way of attacking. 

Regarding the issues above, Kwon Y first proposed the concept of a FAW attack in 2017 [17]. The FAW attack is a new attack that combines the BWH attack and the SM attack, which can allocate the attacker’s computing power to different target mining pools and obtain higher attack revenue. For example, Wang et al. [20] proposed a hybrid attack based on the reinforcement learning method. The attacker uses the reinforcement learning method to switch between the best attack schemes among BWH, FAW, and PAW to maximize the revenue. Ke et al. [21] proposed an improved FAW attack. After the FAW attack is implemented to form a fork, the attack miners will immediately convert from launching the FAW attack to honest mining. The attacker’s revenue is improved by converting between the honest mining strategy and the FAW attack strategy. However, the traditional FAW attack only allocates the attacker’s computing power in a fixed amount, which may result in a waste of computing power, i.e., much computing power on the target mining pool with less revenue. Therefore, some scholars focus on improving the FAW attack to obtain more revenue. Dong et al. [22] proposed a Selfholding attack that is a combined SM attack with a BWH attack, whose attacker adopts the Selfholding attack model to allocate the attack computing power. Gao et al. [23] proposed a computing power adjustment-based interception attack strategy called Power Adjusting Withholding (PAW), which aims to solve the problem of low revenue efficiency caused by the fixed allocation of part of the attack computing power in FAW attacks. If malicious miners dispatched by malicious mining pools find full PoW in multiple target mining pools, when other mining pools find blocks, all malicious miners will submit full PoW to their respective mining pool managers simultaneously. This leads to multiple forks in the network simultaneously, increasing the probability of malicious mining pools gaining revenue. References [20,21,22,23] have improved the distribution of computing power, but they are still unsuitable for using one evaluation mining pool to attack multiple target mining pools. Moreover, these methods cannot adaptively allocate attack computing power according to the size of the target mining pool and lack an evaluation of the effect of attack revenue. In summary, the current BWH and SM attack methods have certain limitations. More specifically, traditional FAW attack methods cannot directly apply to evaluate the scenarios of multiple mining pool attacks and cannot adaptively adjust the distribution of computing resources to maximize revenue.

## 3. Framework Principle

The procedure of data-driven evaluation framework is as follows: The first procedure is Model Building, which is to model and calculate the revenue of an evaluation mining pool, target mining pool and other mining pools. The second procedure is Model Solving, which is to achieve the adaptive distribution of computing power based on the proposed CPAOA. Thus, the framework can help improve the protection capability of the mining pool seriously affected by FAW attacks (weak ability to withstand attacks), and finally it can achieve the security evaluation of mining pools in the blockchain network. 

In this section, we construct a scenario where there are multiple target mining pools in a blockchain network. Then we analyze the revenue of different mining pools in the blockchain network when an attacking mining pool uses FAW to attack multiple target mining pools, and give the following assumptions.

**Assumptions**: The mining pools in the network consist of the evaluation mining pools, target mining pools and other mining pools. Note that other mining pools only conduct honest consensus, and target mining pools help maintain their blockchain system and earn revenue through honest consensus. A part of the consensus computing power in the evaluation mining pool is used for honest consensus to obtain consensus revenue, and the other part of the FAW attack’s computing power is allocated according to the size of the target mining pool’s computing power. Figure 1 shows a schematic diagram of the FAW attack of an evaluation mining pool on multiple target mining pools, in which part of the computing power of the evaluation mining pool is used for its honest consensus, and the other part of the computing power is used for the FAW attack. The evaluation mining pool adjusts and allocates the FAW attack computing power according to the computing power of target mining pools, and launches the FAW attack. Then it reduces the revenue of the target mining pools and improves the average revenue of the evaluation mining pool. Finally, it effectively evaluates the protection capability of the target mining pool.

However, to maximize the revenue of FAW attacks, the following two problems remain to be solved: The first problem is how to use mathematical equations to express the consensus cost and revenue of evaluation mining pools, establish a revenue optimization model for optimal distribution of computing power under FAW attacks, and establish different mining pool revenue models to evaluate the revenue after adaptive FAW attacks. The second problem is how to use the optimization algorithm to solve the problem and obtain the optimal solution for the distribution of attack computing power, thereby improving the evaluation effect of the target mining pool.

### 3.1. Model Building

Note that $\beta_{i}$ represents the computing power of the *i*-th target mining pool, parameter *n* represents the number of target mining pools, $P_{B}=\sum_{i=1}^{n}\beta_{i}$ represents the total computing power of all target mining pools. Parameter α and $P_{O}$ represent the total computing power of evaluation mining pools and other mining pools, respectively. Parameter $P_{T}$ represents the total computing power of the entire network, i.e., $P_{T}=P_{B}+\alpha +P_{O}$. The revenue model is established according to the revenue and consumption cost of the mining pool in [17,23,24]. The revenue of the evaluation mining pool consists of the following three parts:

The first part of the evaluation mining pool’s revenue is the reward for the successful mining by honest consensus of attack computing power [17], which is determined by the proportion of its computing power for honest consensus to the entire network’s consensus computing power, and can be described as follows:(1)$$R_{1}=\frac{(1-\sum_{i=1}^{n}\tau_{i})\alpha }{P_{T}-\sum_{i=1}^{n}\tau_{i}\alpha },$$
where $R_{1}$ represents the revenue of honest consensus by the evaluation mining pool.

The second part of the revenue is the sum of the revenue obtained by the distribution of each computing power of attack $\tau_{i}\alpha$ in the target mining pool when BWH attacks are launched to each target mining pool by computing power of attacks [23]. It is determined by the probability of the target mining pool digging a block and the proportion of revenue, which can be described as follows:(2)$$R_{2}=\sum_{i=1}^{n}\left(\frac{\beta_{i}}{P_{T}-\sum_{i=1}^{n}\tau_{i}\alpha }\times \frac{\tau_{i}\alpha }{\beta_{i}+\tau_{i}\alpha }\right),$$
where $R_{2}$ represents the revenue of BWH attack of the evaluation mining pool.

The rest of the revenue comes from the successful fork. When the illegal miners find the valid block, the computing power of attack (i.e., $\tau_{i}\alpha$) for the *i*-th target mining pool will immediately broadcast the blocks which have been discovered before and successfully generate a fork [23]. In this case, the total revenue obtained by all of the attack computing power form the revenue of successful forks. More specifically, the third revenue is determined by the probability of successful mining, the probability of successful mining by other target mining pools and the proportion of revenue, which can be described as follows:(3)$$R_{3}=\sum_{i=1}^{n}\left(c_{i}\times \frac{\tau_{i}\alpha }{P_{T}}\times \frac{P_{T}-\alpha -\beta_{i}}{P_{T}-\sum_{i=1}^{n}\tau_{i}\alpha }\times \frac{\tau_{i}\alpha }{\beta_{i}+\tau_{i}\alpha }\right),$$
where $R_{3}$ represents the revenue of the successful fork of the evaluation mining pool, and $c_{i}$ represents the probability that the attacking computing power $\tau_{i}\alpha$ submits a block in the *i*-th target mining pool successfully on the chain.

Consider that when evaluation mining pool needs to bear costs such as electricity and equipment loss when conducting honest consensus and FAW attacks, let $C_{h}$ represent the consensus cost per unit of computing power, and $C_{f}$ represents the cost of a FAW attack per unit of computing power, then the cost of attacking the mining pool can be described as follows:(4)$$C=(1-\sum_{i=1}^{n}\tau_{i})\alpha \times C_{h}+\sum_{i=1}^{n}\tau_{i}\alpha \times C_{f}.$$

Thus, the evaluation mining pool’s total revenue $R_{A}$ can be calculated as follows [24]:(5)$$R_{A}=(R_{1}+R_{2}+R_{3}-C)\times R_{Block},$$
where $R_{A}$ sums up the aforementioned three revenues, minus the cost of consumption and multiplied by the reward for mining to a single block $R_{Block}$.

Therefore, the average revenue ${\bar{R}}_{A}$ of the evaluation mining pool’s computing power is the ratio of the total revenue of the evaluation mining pool to the computing power of attack, which can be described as follows:(6)$$\begin{array}{l} {\bar{R}}_{A}=R/\alpha =(R_{1}+R_{2}+R_{3}-C)\times R_{Block}/\alpha \\ \;\;=(\frac{(1-\sum_{i=1}^{n}\tau_{i})\alpha }{P_{T}-\sum_{i=1}^{n}\tau_{i}\alpha }+\sum_{i=1}^{n}\left(\frac{\beta_{i}}{P_{T}-\sum_{i=1}^{n}\tau_{i}\alpha }\times \frac{\tau_{i}\alpha }{\beta_{i}+\tau_{i}\alpha }\right)+\sum_{i=1}^{n}\left(c_{i}\times \frac{\tau_{i}\alpha }{P_{T}}\times \frac{P_{T}-\alpha -\beta_{i}}{P_{T}-\sum_{i=1}^{n}\tau_{i}\alpha }\times \frac{\tau_{i}\alpha }{\beta_{i}+\tau_{i}\alpha }\right)\\ \;\;\;\;-((1-\sum_{i=1}^{n}\tau_{i})\alpha \times C_{h}+\sum_{i=1}^{n}\tau_{i}\alpha \times C_{f}))\times R_{Block}/\alpha . \end{array}$$

The following revenue optimization model for FAW evaluation mining pools can be modeled as follows:(7)$$\max({\bar{R}}_{A}),\;s.t.\;Eqs.\;(1)–(6),\tau_{i}\in [0,1],\forall i,\;\sum_{i=1}^{n}\tau_{i}\leq 1.$$

Similarly, the average revenue of the target mining pool’s computing power is ${\bar{R}}_{I}$, i.e., the ratio of the total revenue of the target mining pool to its total computing power, which can be described as follows:(8)$${\bar{R}}_{I}=(\frac{\sum_{i=1}^{n}\beta_{i}}{P_{T}-\sum_{i=1}^{n}\tau_{i}\alpha }-\sum_{i=1}^{n}\beta_{i}\times C_{h})\times R_{Block}/(\sum_{i=1}^{n}\beta_{i}+\sum_{i=1}^{n}\tau_{i}\alpha ).$$

The average revenue ${\bar{R}}_{O}$ of other mining pool’s computing power is the ratio of the total revenue of other mining pools to their total computing power, which can be described as follows:(9)$${\bar{R}}_{O}=(\frac{P_{O}}{P_{T}-\sum_{i=1}^{n}\tau_{i}\alpha }-P_{O}\times C_{h})\times R_{Block}/P_{O}.$$

### 3.2. Model Solving

The time complexity of directly solving the optimization model (6) by using the optimization theory is relatively high; therefore, a heuristic algorithm is used to solve it. Aquila optimizer algorithm (AO) [25] is one of the latest heuristic algorithms, simulating Aquila’s high-altitude soaring and contour flight. Its globally optimal solution range can be quickly determined, then Aquila’s low-altitude flight and capture can be accurately searched for the optimal solution, which has a good optimal solution search ability. However, the AO algorithm mainly selects different optimization links according to the value of the random number and cannot reach the convergence state in a short time. At the same time, the algorithm does not randomly adjust the solution in the optimization process, and it is easy to fall into the local optimal solution problem. Therefore, this paper proposes a model-solving algorithm based on improved Aquila optimization for the above two problems. That is, by improving the selection mechanism of different optimization links, improving the convergence speed, and combining the idea of scout bees in the artificial bee colony optimization algorithm, the solution is replaced according to the similarity ratio difference of the solution in multiple iterations, to effectively avoid the AO algorithm getting stuck in the local optimal solution and efficiently find the global optimal solution. The details are as follows:

First, the constraints in the model (7) are combined to construct a search space, and *K* attack computing power allocation schemes for evaluation mining pool are randomly generated in the search space; the allocation scheme matrix $X=\{x_{1},x_{2},\ldots ,x_{k},\ldots ,x_{K}\}$ is formed based on this, where $x_{k}$ represents the *k*-th attack computing power distribution scheme. Therefore, the fitness of different allocation schemes can be calculated by the following Equation (10), and the attack computing power allocation scheme with the greatest fitness is selected as the optimal computing power allocation scheme $x_{best}$.
(10)$$\begin{array}{l} F_{k}={\bar{R}}_{A}=(R_{1}+R_{2}+R_{3}-C)\times R_{Block}/\alpha \\ \;\;\;\;\;\;\;=(\frac{(1-\sum_{i=1}^{n}\tau_{i})\alpha }{P_{T}-\sum_{i=1}^{n}\tau_{i}\alpha }+\sum_{i=1}^{n}\left(\frac{\beta_{i}}{P_{T}-\sum_{i=1}^{n}\tau_{i}\alpha }\times \frac{\tau_{i}\alpha }{\beta_{i}+\tau_{i}\alpha }\right)+\sum_{i=1}^{n}\left(c_{i}\times \frac{\tau_{i}\alpha }{P_{T}}\times \frac{P_{T}-\alpha -\beta_{i}}{P_{T}-\sum_{i=1}^{n}\tau_{i}\alpha }\times \frac{\tau_{i}\alpha }{\beta_{i}+\tau_{i}\alpha }\right)\\ \;\;\;\;\;\;\;\;\;-((1-\sum_{i=1}^{n}\tau_{i})\alpha \times C_{h}+\sum_{i=1}^{n}\tau_{i}\alpha \times C_{f}))\times R_{Block}/\alpha ,\;\forall \tau_{i}\in x_{k}, \end{array}$$
where $F_{k}$ represents the fitness value of the *k*-th evaluation mining pool’s computing power distribution scheme. In different iterative processes, it is necessary to adjust the attack computing power distribution scheme for evaluation mining pools. M represents the maximum number of iterations, m represents the current number of iterations, LTh and UTh represent different similarity ratio thresholds, and Op represents the switching parameter in the optimization stage; thus, the specific update of the allocation scheme is as follows:

1. When $m\leq M\times Op\;and\;F_{k}>UTh$, due to the long distance between the current allocation scheme and the optimal allocation scheme, it is necessary to simulate Aquila’s high-altitude soaring through the following Equation (11) to quickly determine the search range of the optimal attack computing power allocation scheme. It is convenient for finding the optimal distribution scheme of attack computing power in the later stage.
(11)$$x_{k,j}^{m+1}=x_{best}\times (1-\frac{m}{M})+(x_{c}-x_{best})\times Rand_{1},\;\forall k,$$
where $x_{k,j}^{m}$ represents the *k*-th attack computing power distribution scheme under the *m*+1-th iteration, $x_{c}$ represents a vector of size ω×1, which is composed of the average value of the current attack computing power distribution scheme matrix *X*, $x_{best}$ represents the optimal allocation scheme in the current iteration process, and $Rand_{1}$ represents a random number in the range from 0 to 1.

2. When $m\leq M\times Op\;and\;F_{k}\leq UTh$, the search scope of the optimal allocation scheme is relatively wide. The scheme uses the following Equation (12) to simulate the situation where the Aquila hovers above the prey. Then it further narrows the current search range by the constant altitude flight of the short glide attack. It is convenient for the quick approach of the optimal attack computing power allocation scheme in the later stage.
(12)$$x_{k,j}^{m+1}=x_{best}\times R_{L}+x_{d}+(\mu_{1}-\mu_{2})\times Rand_{1},\;\forall k,$$
where $R_{L}$ represents a Levy flight distribution function, $x_{d}$ represents the randomly selected attack computing power allocation scheme in the allocation scheme matrix *X*, and $\mu_{1}$ and $\mu_{2}$ represent the vectors of random values for simulating the Aquila Spiral Search.

3. When $m>M\times Op\;and\;F_{k}>LTh$, the attack computing power distribution plan needs to go through the transition from large-scale search to small-scale optimization. The allocation scheme uses Equation (13) to simulate the Aquila to determine the precise area of the prey, using low-altitude flight and rapid attack. Then it updates and quickly approaches the optimal distribution scheme of attack computing power, so that the next link can accurately search for the optimal distribution scheme of attack computing power.
(13)$$x_{k,j}^{m+1}=(x_{best}-x_{mean})\times \omega_{1}-Rand_{2}+((up-lp)\times Rand_{1}+lp)\times \omega_{2},\;\forall k,$$
where $x_{mean}$ represents the mean vector of the current allocation plan matrix *X* in different dimensions, $\omega_{1}$ and $\omega_{2}$ represent allocation scheme search parameters in the range 0 to 1, $Rand_{2}$ represents a vector of size n×1 consisting of random numbers in the range 0 to 1, up represents a vector of size n×1 composed of the maximum values of different dimensions, and lp represents a vector of size n×1 consisting of the minimum values of different dimensions.

4. When $m>M\times Op\;and\;S_{k}\leq LTh$, the attack computing power distribution scheme is close to the optimal solution. The allocation scheme calculates the quality function f to ensure accurate search by Equation (14) and uses Equation (15) to simulate the Aquila accurately grasping and capturing the prey, that is, to find the optimal attack computing power distribution plan.
(14)$$f=m^{\frac{2\times Rand_{1}-1}{{\left(1-M\right)}^{2}}},$$
(15)$$x_{k,j}^{m+1}=f\times x_{best}-(g_{1}\times x_{k}^{m}\times Rand_{1})-g_{2}\times R_{L}+Rand_{2}\times g_{1},\;\forall k,$$
where $g_{1}$ represents the random value in the process of simulating the Aquila capturing prey and $g_{2}$ represents the flight slope in the process of simulating the Aquila capturing the prey.

To avoid the situation that the Aquila loses the globally optimal solution and falls into the local optimal solution due to the limited flight range of the Aquila during the hunting process. When the algorithm falls into a local optimal solution or the attack computing power allocation scheme does not meet the constraints, a new attack computing power allocation scheme is generated by Equation (16) for replacement. Then the local optimal solution is updated to ensure the newly generated attack computing power differences in allocation schemes within the search space.
(16)$${\bar{x}}_{t,j}=\frac{x_{t,j}+(x_{t,j}-x_{r,j})\times Rand_{3}}{\sum_{j=1}^{n}\left(x_{t,j}+(x_{t,j}-x_{r,j})\times Rand_{3}\right)},0<t\leq K,$$
where ${\bar{x}}_{t,j}$ represents the regenerated attack computing power distribution scheme, $x_{t,j}$ represents the attack computing power of the *j*-th target mining pool in the *t*-th attack computing power allocation scheme that needs to be replaced in the allocation scheme matrix, and $x_{r,j}$ represents the randomly selected attack computing power distribution scheme in the domain *r*. $Rand_{3}$ represents a vector of size n×1 consisting of uniformly distributed random numbers in the range −1 to 1. Considering that the total computing power of target mining pools should not be greater than the total computing power of evaluation mining pools, that is, the proportion of attack computing power should not be greater than the sum of 1, the proportion of attack computing power is updated in the solution update.

Repeating the above steps, the evaluation mining pool allocates attack computing power to different target mining pools according to the optimal allocation scheme of attack computing power and launches FAW attacks on target mining pools. By effectively attacking the target mining pool, we can evaluate the target mining pool’s protection capability against FAW attacks, and finally lay the foundation for improving the security of the mining pool.

## 4. Algorithm Implementation

The evaluation mining pool launches FAW attacks on the target mining pool by executing the CPAOA in the evaluation framework, thereby evaluating the protection capability of the target mining pool. The specific pseudo-code is as follows (Algorithm 1).
**Algorithm 1: Computing Power Allocation Optimization Algorithm (CPAOA)**Input: Number of target mining pools1:M = 1000; K = 50; Op = 1/4; Uth = 90; Lth = 80;2:     Calculate the computing power of the target mining pool and other mining pools;3:     Establish the revenue models of evaluation mining pool, target mining pool, and other mining pool;4:     Determine the solution model and construct the search space;5:     for (I = 0; I < T; i++) 6:     Randomly generate K attack computing power distribution schemes and form them into a distribution scheme matrix X;7:     for (j = 0; j < K; j++)8:          Calculate the fitness of each attack computing power distribution scheme;9:          Update the optimal allocation scheme;10:        if judged that the current scheme is in one of the four stages of AO algorithm;11:            Execute the formula at this stage to update the attack computing power distribution scheme;12:            Calculate the fitness value of the scheme;13:            Calculate the fitness value after the artificial bee colony updates the scheme;14:            Compare with the optimal scheme at the current moment and update the optimal scheme;15:         or else 16:            Obtain the optimal distribution scheme of attack computing power for this round;17:            T = T + 1;18:            Update global optimal scheme;19:          end20:     end21:     Determine the optimal distribution scheme of attack computing power;22:     Launch FAW attacks on multiple target mining pools;23:     Evaluate the protection capabilities of multiple target mining pools;

In line 1, the framework initializes parameters, such as the number of iterations M, the number of randomly generated schemes K, the switching parameter Op, the threshold parameter UTh, and threshold parameter LTh. In line 2, the evaluation mining pool calculates the computing power of the target mining pool and other mining pools. In lines 3–4, the algorithm establishes the models for revenue of evaluation mining pool, target mining pool, and other mining pool. At the same time, considering the attack cost and other models, the final determines the solution model and constructs the search space. In line 6, the algorithm randomly generates K attack computing power distribution schemes and forms them into a distribution scheme matrix $X=\{x_{1},x_{2},\ldots ,x_{k},\ldots ,x_{K}\}$. In lines 7–21, the algorithm continuously updates the scheme through the improved AO algorithm. According to the scheme being in one of the four stages, which are Aquila’s high-altitude soaring, contour flight, low-altitude flight and capture, the algorithm calculates the fitness value of each scheme. After multiple rounds of iterations, the algorithm selects the optimal distribution scheme of attack computing power and records the allocated computing power of each target mining pool. In lines 22–23, according to the optimal distribution scheme of attack computing power, the evaluation mining pool launches FAW attacks on multiple target mining pools according to the optimal distribution scheme of attack computing power. Thus, CPAOA evaluates the protection capabilities of multiple target mining pools.

## 5. Experiment and Analysis

### 5.1. Parameters and Performance Indicators

To test the performance of CPAOA, we build a prototype blockchain system based on PoW consensus on a server with Intel i7-10700 CPU 2.90GHz and 16G memory, and the operating system is 64-bit Win10. We calculate the revenue value of each mining pool in our proposed algorithm. Using the experimental parameters in Table 1, we calculate the optimal distribution scheme of attack computing power and the computing power ratio in the target mining pool, the average revenue of the evaluation mining pool, and the average revenue of the target mining pool, respectively. We also consider that the total computing power of target mining pools may change and compare the revenue of CPAOA, BWH, SM and FAW_F. Note that FAW_F is an attack strategy that no matter how the computing power of the target mining pool changes, the evaluation mining pool always maintains a fixed distribution ratio of computing power to launch FAW attacks.

### 5.2. Parameter Selection Analysis

In order to analyze the influence of the fitness threshold parameter on the number of iterations when the algorithm reaches the optimal solution, let the fitness threshold parameter UTh be 50, 70, 90, 110 and 130, respectively. The parameters LTh are set as 60, 80, 100, 120, and 140, respectively. The given number of iterations is 1000, and the number of target pools is 9. Figure 2 shows the parameter diagram of Aquila optimization. As shown in Figure 2, the fitness threshold parameter can affect the range of solution optimization in the CPAOA optimization process. When the parameter UTh increases continuously in the range of 50 to 90, CPAOA is in the large-scale search stage of soaring at high altitude during the optimization process. This is, the larger search range of the algorithm, the shorter search time in reaching the vicinity of the optimal solution. Therefore, the number of iterations when the algorithm reaches the optimal solution decreases gradually. However, when the parameter UTh increases continuously in the range of 90 to 130, CPAOA is in the large-scale search stage of hovering and gliding during the optimization process. At this time, a lot of iteration rounds will lead to an excess of search capacity, and the optimization of the low-altitude flight and dive capture stage will start late, so the number of iterations when the algorithm reaches the optimal solution gradually increases. When the parameter LTh increases continuously in the range of 60 to 80, CPAOA is in the low-altitude flight stage during the optimization process, the search range increases, and the algorithm can accurately search for the optimal solution. As a result, the number of iterations when the algorithm reaches the optimal solution gradually decreases. However, when the parameter LTh increases continuously in the range of 80 to 140, CPAOA is in the stage of small-scale diving to capture prey during the optimization process, and the increase of the search range will make it difficult to obtain the optimal solution quickly and accurately. Therefore, the number of iterations when the algorithm reaches the optimal solution gradually increases. Through parameter analysis, it can be concluded that when CPAOA selects UTh as 90 and LTh as 80, CPAOA can better allocate the four-stage range of optimization to find the optimal solution quickly.

The switching parameter Op in the optimization stage are selected as 1/6, 1/4 and 1/2, respectively. The fitness threshold parameter UTh is 90, the parameter LTh is 80, the number of iterations is 1000, the number of target mining pools is 9, and other parameters are shown in Table 1. We analyze the influence of switching parameters in the optimization stage on the average revenue of the evaluation mining pool (note that all average revenue in the following refers to the average revenue of computing power in the mining pool). As shown in Figure 3, the switching parameters in the optimization stage will affect the allocation ratio of the large-scale optimization of high-altitude flight and the small-scale optimization of low-altitude capture in the CPAOA optimization process. When the switching parameter in the optimization stage is 1/6, the search times of high-altitude soaring and hovering gliding in the CPAOA optimization process are too short. At this time, the optimization relies on low-altitude flight and dive capture with a small search range, and it is not easy to approach the optimal solution quickly. When the switching parameter in the optimization stage is 1/4, the search range of high-altitude soaring, hovering gliding, low-altitude flight and diving capture in the CPAOA optimization process all reach the appropriate range so that the optimal solution can be obtained quickly. When the switching parameter in the optimization stage is 1/2, the search times of high-altitude soaring and hovering gliding are too many in the CPAOA optimization process, which can quickly approach the vicinity of the optimal solution. However, it needs to wait for the number of iterations to reach 500 before the reliable low-altitude flight and dive capture gradually approach the optimal solution. Therefore, when the switching parameter of the CPAOA optimization stage is selected as 1/4, the optimization iterative search range can be better allocated to find the optimal solution quickly.

The switching parameter Op in the optimization stage is selected as 1/4, the fitness threshold parameter UTh is 90, the parameter LTh is 80, the number of iterations is 1000, the number of target mining pools is 9, and other parameters are shown in Table 1. We analyze the influence of adding the artificial bee colony update mechanism on optimization. As shown in Figure 4, the addition of the artificial bee colony update mechanism can help achieve the optimal fitness value more quickly and accurately, and the function also can converge faster to obtain the optimal scheme of attack computing power distribution. Due to certain constraints in the distribution scheme of attack computing power, the attack computing power allocated to each target mining pool is greater than 0 and less than the computing power of the evaluation mining pools. In the process of continuous iterative optimization by CPAOA, the artificial bee colony update mechanism can modify and update the solution according to the similar ratio difference of the solution in multiple iterations; it can effectively avoid the AO algorithm getting stuck at the local optimal solution and search for the global optimal solution.

### 5.3. Algorithm Performance Analysis

Let ε denote the proportion of the evaluation mining pool’s attack computing power to its total computing power, i.e., $\varepsilon =\sum_{i=1}^{n}\tau_{i}\alpha /\alpha$. The number of target mining pools is 9, the fitness threshold parameter UTh is 90, the parameter LTh is 80, the computing power of target mining pools is 400, and other parameters are shown in Table 1. We analyze and compare the average revenue of a mining pool unit when ε is set to 10%, 20%, 30%, 40%, 50%, 60%, 70%, 80%, 90% and 100%, respectively. As shown in Figure 5, CPAOA can reduce the average revenue of the target mining pool unit’s computing power by 27.54% on average, increase the average revenue of target mining pool unit’s computing power by 22.21% and the average revenue of the other mining pools by 18.87%. CPAOA has a significant effect on illegal behavior containment and is protective for other mining pools. At the same time, when the proportion of attacking computing power reaches 70%, the fluctuation of the average computing power of evaluation mining pools and target mining pools has become stable. As a result, evaluation mining pools get the most revenue by mining honestly and attacking target mining pools. As the proportion of attack computing power allocated to evaluation mining pools increases, the computing power allocated to FAW attacks increases. Accordingly, the honest mining computing power of evaluation mining pools will decrease. It can be seen from the revenue model that the reduction of the honest mining computing power of the entire network will increase the successful mining probability of other mining pools. At the same time, due to the increase in attack computing power, the mining probability of target mining pools decreases, and the FAW attack revenue obtained by evaluation mining pools through $R_{1}$, $R_{2}$ and $R_{3}$ increases. In the process of computing power allocation of the FAW attack, CPAOA finds the optimal solution by improving the selection mechanism of the four optimization stages to improve the convergence speed of the solution of the attack revenue optimization model. Moreover, using the artificial bee colony idea to update the solution can avoid the situation that the attack computing power allocation scheme falls into the local optimal solution or does not meet the constraints. In this way, the optimal distribution scheme of attack computing power can be obtained, and the revenue of attacking the mining pool can be maximized.

According to the results in Figure 6, the selected number of target mining pools is 9, the total computing power of the evaluation mining pool is 500, the computing power of target mining pools is 400, the fitness threshold parameter UTh is 90, the parameter LTh is 80, and other parameters are shown in Table 1. We analyze the computing power of each target mining pool and the distribution of the evaluation mining pool’s computing power. As shown in Figure 6, the distribution of the attack computing power can adaptively allocate the FAW attack computing power according to the computing power of the target mining pool, which maximizes the revenue of the evaluation mining pool. As a result, CPAOA establishes revenue models for different mining pools and improves the calculation method of fitness value in the optimization process. At the same time, given the shortcomings of the traditional AO optimization solution, which makes it easy to fall into the local optimal solution without adjustment, CPAOA improves the selection mechanism of different optimization links, combined with the idea of the scout bee in the artificial bee colony optimization algorithm. CPAOA updates the solution in time to meet the constraints to avoid falling into a local optimal solution. Finally, CPAOA obtains the globally optimal solution and allocates the computing power in a form suitable for the computing power of each target mining pool to maximize the revenue of evaluation mining pools.

The number of target mining pools is 9, the fitness threshold parameter UTh is 90, the parameter LTh is 80, and other parameters are shown in Table 1. We analyze the average revenue of the target mining pool’s computing power when the total computing power of the target mining pool and the evaluation mining pool is 200, 300, 400, 500, and 600, respectively. As shown in Figure 7, with the computing power of evaluation mining pools increasing, the average computing power of all target mining pools can be effectively suppressed and has a downward trend. With the increase in the computing power of target mining pools, the average revenue of target mining pools has increased. However, the increase has been effectively restrained, narrowing the trend. As a result, when the computing power of the target mining pool is constant, with the computing power of the evaluation mining pool increasing, the computing power of mining in the whole network increases, and the FAW attack computing power of the evaluation mining pool to the target mining pool also increases. Therefore, the probability of target mining pools being calculated as mines is reduced, and more revenue needs to be allocated to the FAW attack computing power, decreasing the average revenue of target mining pools. When the computing power of the evaluation mining pool is constant, with the computing power of the target mining pool increasing, although the probability of target mining in the target mining pool increases, the CPAOA will calculate the computing power by attacking the mining pool according to the computing power of different target mining pools. The power revenue optimization model adaptively allocates attack computing power, launches effective FAW attacks on multiple target mining pools by a single mining pool, and obtains more target mining pool revenue. Therefore, as the computing power of target mining pools increases, the increase in target mining pool income tends to narrow, which is effectively suppressed.

### 5.4. Algorithm Comparison Analysis

The number of target mining pools is 9, the computing power of target mining pools is 400, the fitness threshold parameter UTh is 90, the parameter LTh is 80, and other parameters are shown in Table 1. We analyze and compare the average revenue of evaluation mining pools and target mining pools for CPAOA, BWH, SM, and FAW_F when the computing power of evaluation mining pools is set to 200, 300, 400, 500, and 600, respectively. As shown in Figure 8 and Figure 9, no matter how the computing power of the evaluation mining pool changes, the average computing power of the evaluation mining pools of CPAOA is higher than the average computing power of the evaluation mining pools of BWH, SM and FAW_F, which is increased by at least 17.82%. The average hash rate returns of the target mining pools are lower than those of BWH, SM and FAW_F, and they are reduced by at least 20.04%. As a result, CPAOA establishes an optimization model for the revenue of the evaluation mining pool’s computing power, based on four-stage optimization and local optimal solution updating. This iterates continuously based on the fitness value in the established revenue model. It obtains the optimal solution and thus obtains the optimal solution for different target mines. To maximize the average computing power of evaluation mining pools and minimize the average computing power of target mining pools, CPAOA has the highest average computing power of evaluation mining pools and the lowest average computing power of target mining pools. However, FAW_F always maintains a fixed distribution of attack computing power and cannot cope with the increase in the total computing power of the entire network and the uneven allocation of computing power in target mining pools; thus, the average revenue of its evaluation mining pools is lower than CPAOA. At the same time, with the computing power of the attacked mining pools increasing, FAW_F will gradually approach the optimal computing power distribution. The average computing power of target mining pools has a great downward trend, but it is always higher than CPAOA. Computing power of BWH attack only gains revenue by reporting part of the proof of work in the target mining pool. Computing power of SM attack chooses to broadcast the calculated mine immediately when other miners calculate the mine. Both BWH and SM are only part of the FAW attack. Therefore, the average computing power of evaluation mining pools under BWH and SM is lower than that of FAW attack, and the average computing power of target mining pools is higher than that of CPAOA.

We also analyze the experimental data and calculate the total average revenue of CPAOA, BWH, SM and FAW_F, that is the average revenue of mining pool under different computing power (i.e., 200, 300, 400, 500 and 600). Table 2 shows the comparative results of total average revenue with different algorithms. In Table 2, with respect to the evaluation mining pools, CPAOA is 30.54% higher than BWH on average, 186% higher than SM, and 17.82% higher than FAW_F. With respect to the target mining pools, CPAOA is 38.83% lower than BWH, 20.04% lower than SM, and 32.07% lower than FAW_F. The above shows that CPAOA can more effectively allocate the computing power of the evaluation mining pools. CPAOA can also evaluate the target mining pools more efficiently and improve the security of the target mining pool with weak protection capability evaluated. The data-driven evaluation framework can ensure the revenue and security of the mining pools and participants in the blockchain network.

## 6. Conclusions

In this paper, which considers the effective evaluation of the target mining pool’s protection capability against FAW attacks, we propose a novel evaluation framework for FAW attack protection of the target mining pools. In this framework, aiming at the evaluation of multiple target mining pools, we propose a computing power allocation optimization algorithm (CPAOA) for FAW attack mining pools. Firstly, the framework proposes the honest consensus revenue, block withholding revenue, successful fork revenue, and consensus cost with mathematical equations, and establishes the revenue model of different mining pools to evaluate the attack revenue. Then, CPAOA transforms the adaptive allocation of the computing power of evaluation mining pools into the optimization problem of the revenue optimization model. Moreover, CPAOA proposes an improved solution algorithm. The algorithm is based on the Aquila optimization method to calculate the optimal solution of the revenue model by improving the selection mechanism of different optimization processes. Then CPAOA proposes the solution update mechanism of the local optimal solution, which can improve the efficiency of optimization and reduce the probability that the algorithm will fall into the local optimal solution. Based on the experimental analysis, we analyze the influence of the CPAOA algorithm’s parameters on its performance. Compared with traditional attack methods, CPAOA can significantly improve the average revenue of evaluation mining pools and reduce the average revenue of the target mining pools. The proposed framework can efficiently evaluate the protection capability of the target pool against FAW attacks.

In future works, we will extend the evaluation framework to more consistency attacks and consider the case of multiple-to-multiple mining pools. The framework will provide appropriate measures for mining pools, thus further improving the security of blockchain mining pools.

## Figures and Tables

**Figure 1 sensors-22-09125-f001:**
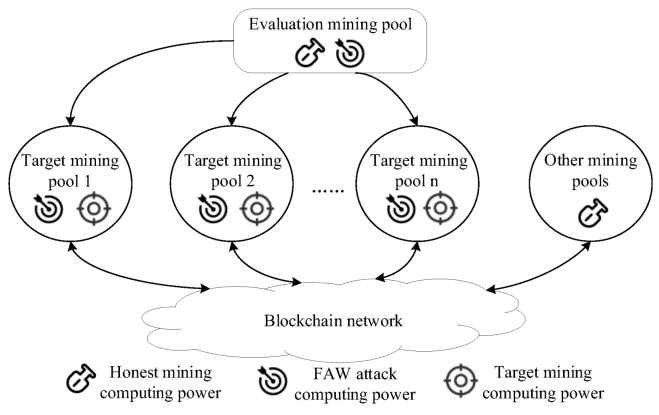
Schematic diagram of FAW attack against multiple target mining pools.

**Figure 2 sensors-22-09125-f002:**
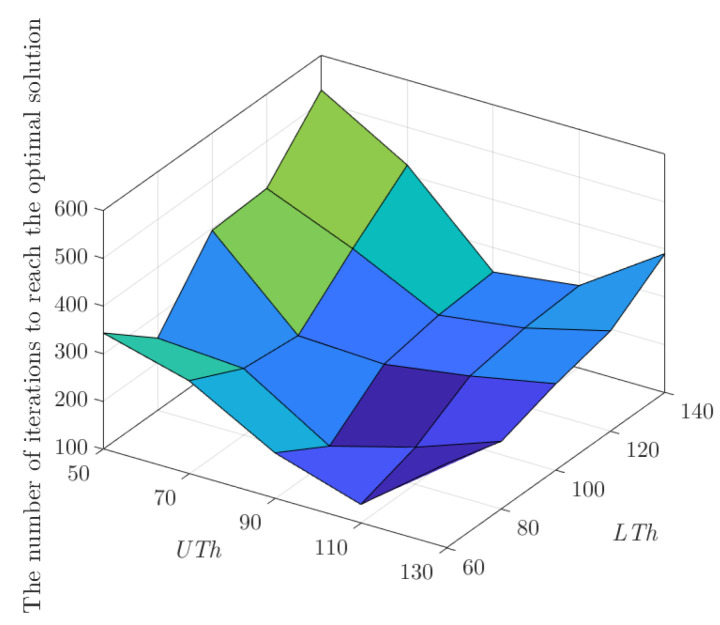
Aquila optimization parameter diagram.

**Figure 3 sensors-22-09125-f003:**
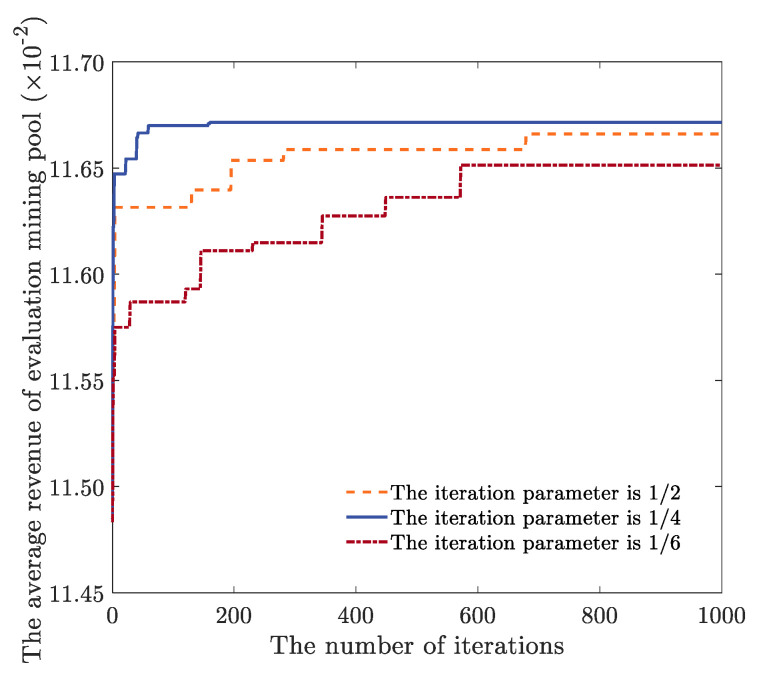
Optimization stage switching parameter diagram.

**Figure 4 sensors-22-09125-f004:**
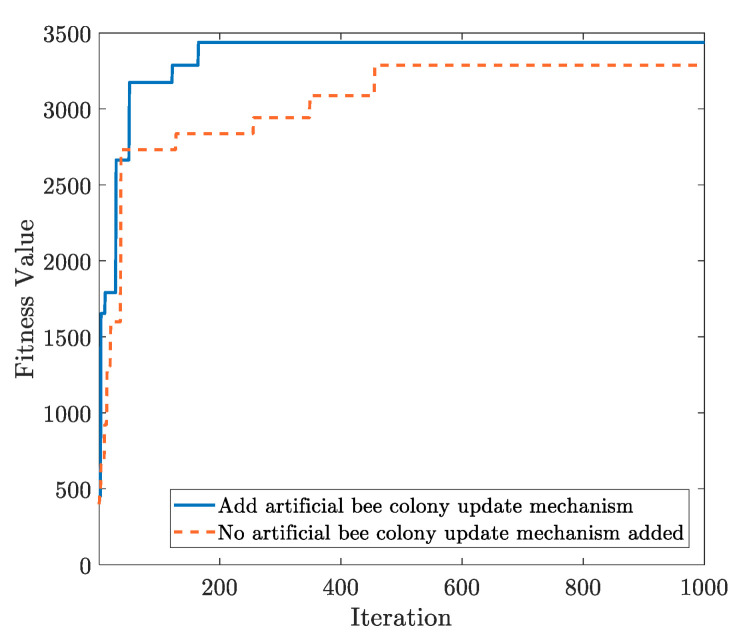
Comparison of artificial bee colony update mechanism.

**Figure 5 sensors-22-09125-f005:**
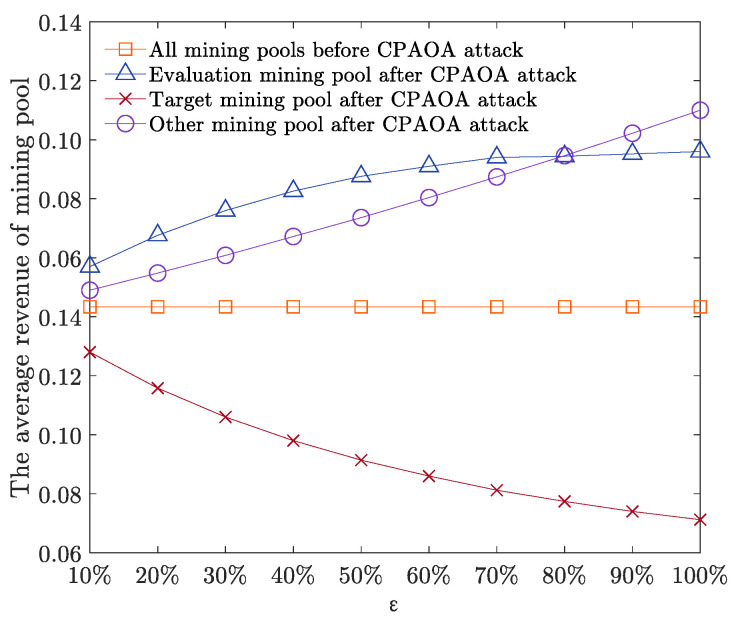
The average revenue of mining pool computing power before and after the attack.

**Figure 6 sensors-22-09125-f006:**
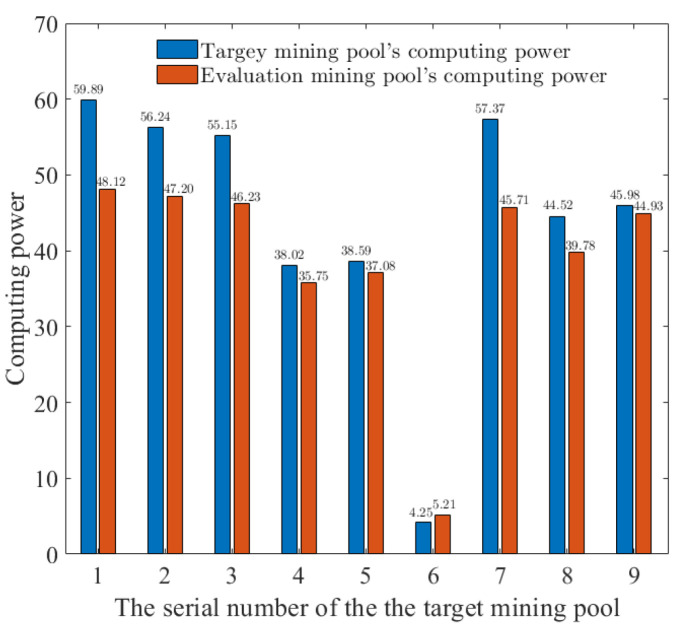
Distribution of computing power for FAW attacks on multiple target mining pools.

**Figure 7 sensors-22-09125-f007:**
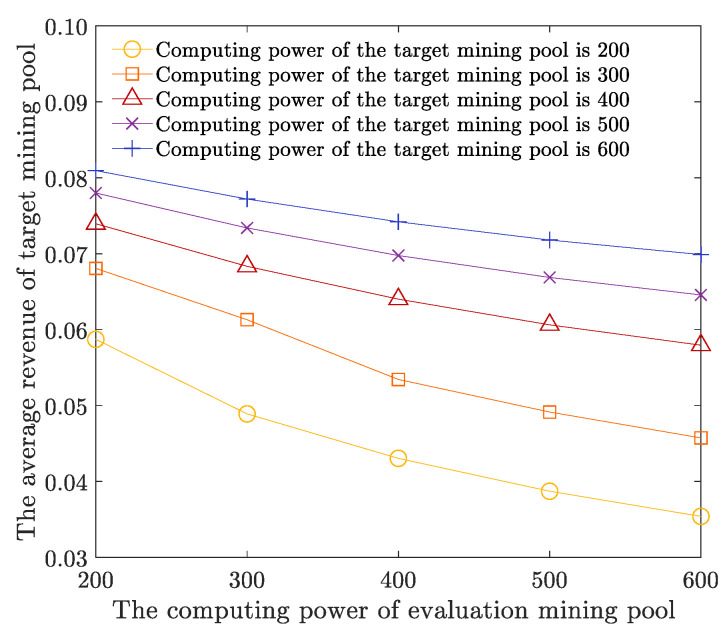
Average revenue of evaluation mining pools and target mining pools with different computing power.

**Figure 8 sensors-22-09125-f008:**
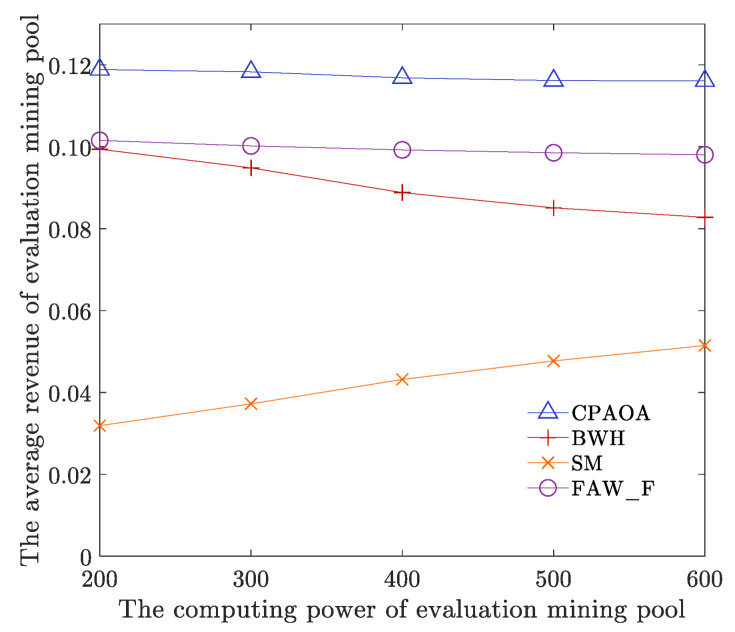
Comparison of the average revenue of algorithms with computing power of an evaluation mining pool.

**Figure 9 sensors-22-09125-f009:**
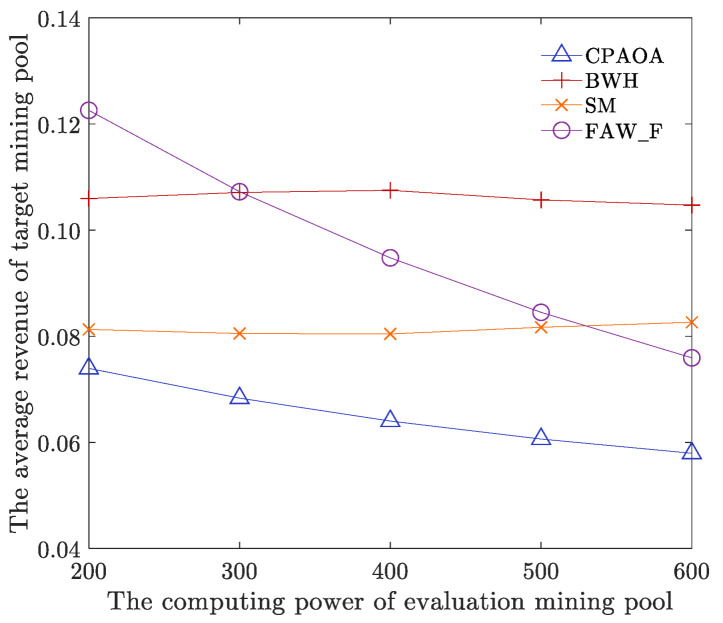
Comparison of the average revenue of algorithms with computing power of a target mining pool.

**Table 1 sensors-22-09125-t001:** Experimental parameter table.

Parameter	Number
Number of iterations M	1000
Number of randomly generated evaluation mining pool computing Power distribution schemes K	50
Consensus cost per unit of computing power $C_{h}$	0.00015
Cost of FAW attack per unit of computing power $C_{f}$	0.00005
Computing power of the entire network $P_{T}$	3000
Probability of successful hashing of blocks in target mining pools c	0.6
Reward of single block consensus $R_{Block}$	500

**Table 2 sensors-22-09125-t002:** Experimental summary table.

Algorithm	Mining Pool	Total Average Revenue	Comparison
CPAOA	Evaluation mining pool	0.1173	1
	Target mining pool	0.0650	1
BWH	Evaluation mining pool	0.0902	30.54%
	Target mining pool	0.1062	−38.83%
SM	Evaluation mining pool	0.0423	186.04%
	Target mining pool	0.0813	−20.04%
FAW_F	Evaluation mining pool	0.0995	17.82%
	Target mining pool	0.0970	−32.07%

## Data Availability

Not applicable.

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
