# Peer review of "A Novel Data-Driven Evaluation Framework for Fork after Withholding Attack in Blockchain Systems"

_sensors, 2022, doi:10.3390/s22239125_

Round 1

Reviewer 1 Report

The following comments are required to update in the paper

-        Avoid using acronyms in the title. Make title understandable

-        this paper proposes a novel evaluation framework for FAW attack 17 protection of the target mining pools in blockchain systems using “computing power allocation optimization algorithm (CPAOA)”

-        what are the basic criteria and validation of establishing the revenue model for mining pools in this study?

-        What are the other mining pools other than target mining pools in this paper? Kindly list all.

-        In figure 5, kindly show the value of each bar.

K   kindly include a proposed approach which can show the step-wise-step procedure of whole work done in this paper.

Author Response

Thanks for your comments,  Please refer to the attachment for detailed response.

Reviewer 2 Report

Summary
Authors in this paper propose a novel evaluation framework for Fork After
Withholding (FAW) attacks the protection of the target mining pools in blockchain systems. They propose an efficient computing power allocation
optimization algorithm (CPAOA) for FAW attacks against multiple target
mining pools in this framework.
- Strengths
ï‚· The title is appropriate as it is short, clear, and understandable.
ï‚· The main contributions in the manuscript were specified clearly in
the introduction section.
ï‚· They provided enough overview of all terminologies that are
mentioned in this manuscript.
- I have the following comments
ï‚· They must demonstrate more experiments to justify the
effectiveness of their work.
ï‚· They must apply some statistics to all of the simulation results to
show their significance.
ï‚· If the formulation of mathematical modeling (Equ (1) - Equ (16)) are
from previous work, you must cite the paper.

ï‚· Finally, I believe that this paper is good but it suffers from some
problems that must be solved. Good revision is needed before it can
be considered for publication in this journal.

Author Response

Reviewer#2:

Summary

Authors in this paper propose a novel evaluation framework for Fork After Withholding (FAW) attacks the protection of the target mining pools in blockchain systems. They propose an efficient computing power allocation optimization algorithm (CPAOA) for FAW attacks against multiple target mining pools in this framework.

- Strengths

ï‚· The title is appropriate as it is short, clear, and understandable.

ï‚· The main contributions in the manuscript were specified clearly in the introduction section.

ï‚· They provided enough overview of all terminologies that are mentioned in this manuscript.

Author response: Thank the expert for affirming our paper.

Author action: Thank the expert for affirming our paper.

Reviewer#2, Concern # 1:

They must demonstrate more experiments to justify the effectiveness of their work.

Author response: Thanks for the suggestion. We have modified the experiment and analysis in Section 5, specifically including “Parameters and Performance Indicators”, “Parameter Selection Analysis”, “Algorithm Performance Analysis” and “Algorithm Comparison Analysis”. We have added a new experimental process whose results are shown in Figure 4. By comparing experimental results with and without the artificial colony renewal mechanism, the effectiveness of our work can be further reflected. We also have added Table 2 and the theoretical analysis, the specific modification is shown in concern 2. The specific modification of new experiment is as follows.

Figure 4. Comparison of artificial bee colony update mechanism.

The switching parameter  in the optimization stage are selected as 1/4, the fitness threshold parameter  is 90, the parameter  is 80, the number of iterations is 1000, the number of target mining pools is 9, and other parameters are shown in Table 1. We analyze the influence of adding the artificial bee colony update mechanism on optimization. As shown in Fig. 4, the addition of artificial bee colony update mechanism can help achieve the optimal fitness value more quickly and accurately, and the function also can converge faster to obtain the optimal scheme of attack computing power distribution. Due to certain constraints in the distribution scheme of attack computing power, the attack computing power allocated to each target mining pool is greater than 0 and less than the computing power of the evaluation mining pools. In the process of continuous iterative optimization by CPAOA, artificial bee colony update mechanism can modify and update the solution according to the similar ratio difference of the solution in multiple iterations, it can effectively avoid the AO algorithm getting stuck the local optimal solution and search for the global optimal solution.

Author action: According to the reviewer's comment, we have added a new experimental process whose results are shown in Fig. 4 as well as theoretical analysis in the revised manuscript. The specific modification is as mentioned above.

Reviewer#2, Concern # 2:

They must apply some statistics to all of the simulation results to show their significance.

Author response: Thanks for the suggestion. We have summarized the data of Algorithm Performance Analysis and Algorithm Comparison Analysis, and added a new Table 2 to more effectively express the significance of the proposed algorithm. The specific modification is as follows.

  Table 2. Experimental summary table.

Algorithm

Mining pool

Total average revenue

Comparison

CPAOA

Evaluation mining pool

0.1173

1

Target mining pool

0.0650

1

BWH

Evaluation mining pool

0.0902

30.54%

Target mining pool

0.1062

-38.83%

SM

Evaluation mining pool

0.0423

186.04%

Target mining pool

0.0813

-20.04%

FAW_F

Evaluation mining pool

0.0995

17.82%

Target mining pool

0.0970

-32.07%

We also analyze the experimental data and calculate the total average revenue of CPAOA, BWH, SM and FAW_F, that is the average revenue of mining pool under different computing power (i.e., 200, 300, 400, 500 and 600). Table 2 shows the comparation results of total average revenue with different algorithms. In Table 2, in the respect of the evaluation mining pools, CPAOA is 30.54% higher than BWH on average, 186% higher than SM, and 17.82% higher than FAW_F. In the respect of the target mining pools, CPAOA is 38.83% lower than BWH, 20.04% lower than SM, and 32.07% lower than FAW_F. The above shows that CPAOA can more effectively allocate the computing power of the evaluation mining pools. CPAOA can also evaluate the target mining pools more efficiently and improve the security of the target mining pool with weak protection capability evaluated. The data-driven evaluation framework can ensure the revenue and security of the mining pools and participants in the blockchain network.

Author action: Thanks for the suggestion. We have added the Table 2 and theoretical analysis by revising the relevant content of the Algorithm Comparison Analysis in the revised manuscript. For details, please refer to the revised manuscript.

Reviewer#2, Concern # 3:

If the formulation of mathematical modeling (Equ (1) - Equ (16)) are from previous work, you must cite the paper.

Author response: Thanks for the suggestion. We have checked the formula and cited the references of relevant work. The specific modification is as follows.

The revenue model is established according to the revenue and consumption cost of the mining pool in [16,22,23]. The revenue of the evaluation mining pool consists of the following three parts.

The first part of the evaluation mining pool’s revenue is the reward for the successful mining by honest consensus of attack computing power [16].

BWH attacks are launched to each target mining pool by computing power of attacks [22].

Thus, the evaluation mining pool's total revenue  can be calculated as follows [23].

[23] Yang, R.; Chang, X.; Mišić, J.; Mišić, V.; Zhu, H. Evaluating fork after withholding (FAW) attack in Bitcoin. In Proceedings of the 19th ACM International Conference on Computing Frontiers; ACM: New York, NY, USA, 2022; pp. 67-74.

Author action: Thanks for the suggestion. We have added the references [16,22,23]. For details, please refer to the red words section in the “Model Description” in the revised manuscript.

Finally, I believe that this paper is good but it suffers from some problems that must be solved. Good revision is needed before it can be considered for publication in this journal.

Author response: Thank the expert for affirming our paper. We have carefully checked the paper and revised them according to the expert's opinion.

Author action: Thanks for the suggestion. We have carefully checked the paper and revised them according to the expert's opinion.

Round 2

Reviewer 2 Report

The authors did all the required modifications.